# Cucumber Leaf Diseases Recognition Using Multi Level Deep Entropy-ELM Feature Selection

**Muhammad Attique Khan** [1,*] , **Abdullah Alqahtani** [2], **Aimal Khan** [3], **Shtwai Alsubai** [2], **Adel Binbusayyis** [2], **M Munawwar Iqbal Ch** [4], **Hwan-Seung Yong** [5] and **Jaehyuk Cha** [6]

[1] Department of Computer Science, HITEC University Taxila, Taxila 47080, Pakistan
[2] College of Computer Engineering and Sciences, Prince Sattam bin Abdulaziz University, Al-Kharj 16273, Saudi Arabia; Aq.alqahtani@psau.edu.sa (A.A.); Sa.alsubai@psau.edu.sa (S.A.); a.binbusayyis@psau.edu.sa (A.B.)
[3] Department of Computer & Software Engineering, CEME NUST Rawalpindi, Rawalpindi 46000, Pakistan; aimalkhan.eme@gmail.com
[4] Institute of Information Technology, Quaid-i-Azam University, Islamabad 44000, Pakistan; mmic@qau.edu.pk
[5] Department of Computer Science & Engineering, Ewha Womans University, Seoul 03760, Korea; hsyong@ewha.ac.kr
[6] Department of Computer Science, Hanyang University, Seoul 04763, Korea; chajh@hanyang.ac.kr
* Correspondence: attique.khan@hitecuni.edu.pk

**Abstract:** Agriculture has becomes an immense area of research and is ascertained as a key element in the area of computer vision. In the agriculture field, image processing acts as a primary part. Cucumber is an important vegetable and its production in Pakistan is higher as compared to the other vegetables because of its use in salads. However, the diseases of cucumber such as Angular leaf spot, Anthracnose, blight, Downy mildew, and powdery mildew widely decrease the quality and quantity. Lately, numerous methods have been proposed for the identification and classification of diseases. Early detection and then treatment of the diseases in plants is important to prevent the crop from a disastrous decrease in yields. Many classification techniques have been proposed but still, they are facing some challenges such as noise, redundant features, and extraction of relevant features. In this work, an automated framework is proposed using deep learning and best feature selection for cucumber leaf diseases classification. In the proposed framework, initially, an augmentation technique is applied to the original images by creating more training data from existing samples and handling the problem of the imbalanced dataset. Then two different phases are utilized. In the first phase, fine-tuned four pre-trained models and select the best of them based on the accuracy. Features are extracted from the selected fine-tuned model and refined through the Entropy-ELM technique. In the second phase, fused the features of all four fine-tuned models and apply the Entropy-ELM technique, and finally fused with phase 1 selected feature. Finally, the fused features are recognized using machine learning classifiers for the final classification. The experimental process is conducted on five different datasets. On these datasets, the best-achieved accuracy is 98.4%. The proposed framework is evaluated on each step and also compared with some recent techniques. The comparison with some recent techniques showed that the proposed method obtained an improved performance.

**Keywords:** crops diseases; data augmentation; deep learning; entropy; features fusion; machine learning

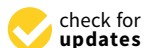



## 1. Introduction

Agriculture is one of the most important research topics globally nowadays [1]. Agriculture is a significant source of income and the economy of a country is based on the quality and yields of crops [2]. Cucumber is an important vegetable and during the year 2020, the global cucumber planting area was around 2.25 million hectares and the global

yield was 90.35 million tons [3]. The production of crops is highly threatened by diseases and failure to identify and prevent cucumber diseases causes a reduction of cucumber vegetable yield and quality. The failure in early diagnosis causes significant economic losses to growers. Therefore, the rapid diagnosis of crops diseases helps to increase the quality and yield and also increases the national economy [4].

Mostly, identification is accomplished using typical methods like seeing through naked eyes or through a microscope [5]. The results of the manual visual estimation are generally unreliable while the microscopic assessments are generally time-consuming and costly. Most of the agriculturists of underdeveloped countries are illiterate [6]. They are compelled to return those charges along with other expenses like pesticides and fertilizer. The cucumber diseases like anthracnose, powdery mildew, downy mildew, and cucumber mosaic can destroy a large number of crops and the result will be a huge loss and vegetable deficiency [7]. Significant work has been done to accomplish a method that can boost the fastness and accuracy of the process. The methods necessarily contained some sort of computerization [8]. A large number of techniques presented until now are based on digital image processing and machine learning to identify the crops' diseases and achieve the desired output [9].

Image processing has many applications in the domain of computer vision such as medical imaging [10], agriculture [11], and named a few more. Agriculture is a hot application of image processing for the identification and classification of crops and plant diseases [12]. Although detection of cucumber abnormalities and then classifying them using image processing techniques is a critical task due to some sequence of steps [13]. A computerized method consists of some important steps such as preprocessing of original leaf images, detection of the infected region, feature extraction using handcrafted methods, and finally reduction and classification. Recognition of diseased portions in images is the key factor as it can influence the design and performance of the classification algorithms [14]. However, the error in the detection of the infected region extracted the irrelevant features that reduces the recognition accuracy.

Deep learning (DL) [15] is a hot research topic nowadays [16] and is employed everywhere for the detection and recognition tasks for several applications [17] such as biometric [18], image classification [19], surveillance [20], medical [21], and agriculture [22]. The researcher of computer vision introduced many techniques using machine learning and deep learning for plants diseases recognition [23]. Hussain et al. [24] introduced a deep learning technique for the identification of multiple cucumber leaf diseases. They extract deep learning features through two fine-tuned deep models including VGG19 and Inception V3. Both fine-tuned models were trained on the selected dataset using the transfer learning approach. The main advantage of training through TL is to save memory and time. The extracted features were fused by implementing the parallel maximum fusion technique to get the maximum information of each trained image. In the end, a whale optimization algorithm (WOA) was applied to select the robust features and perform classification. The purpose of feature selection is to get the best features because, in the fusion process, a few redundant features were also added. They achieved a maximum of 96.5% accuracy on the selected leaf dataset. In [13], researchers built an automated detection classification model for cucumber leaf diseases. In the first phase, pre-processing was performed to enhance the local contrast of images and to make the infected region more visible. This step makes the infected region clearer that later helped in the accurate segmentation using a novel Sharif saliency-based (SHSB) technique. Then researchers fused the proposed saliency method with active contour segmentation to improve the segmentation accuracy that later extracts the relevant features. In the feature extraction phase, they utilized VGG-19 and VGG-M pre-trained models. The extracted features were refined through three parameters including local entropy, local interquartile range, and local standard deviation. In the final classification, the best accuracy of 98.08% was achieved on multi-class SVM. The strength of this work was less computational time that can be useful for a real-time computerized system. Wang et al. [3] introduced a deep learning-based technique for the recognition of

cucumber leaf diseases under complex backgrounds. They fused DeepLabV3+ and U-Net models instead of a single network. In the first step, DeepLabV3+ was used to segment the leaves from the images. Then the diseased area was segmented using U-Net. The fused models give better accuracy than the accuracy reported by the individual models. Researchers in [25], introduced a model for the identification of crop diseases in real-world images. The proposed trilinear convolutional neural network utilized bilinear pooling. In the laboratory environment, the proposed technique achieved 99.99% accuracy and in the real-world environment, the obtained accuracy is 84.11%. Kianat et al. [7] proposed a hybrid system for the recognition of cucumber diseases. In the pre-processing step, the data augmentation was applied using different angles to increase the image count in the dataset. In this step, contrast stretching was also performed to visually improve the images. The features were extracted from binary robust invariant scalable keypoints (BRISK), histogram of gradient (HOG), and features from the accelerated segmented test (FAST). Initially, the irrelevant features were eliminated by utilizing the probability distribution-based entropy (PDbE) technique. Then features were fused using the serial-based method and implemented Manhattan distance-controlled entropy (MDcE) method was to select the robust features. The proposed model achieved maximum accuracy of 93.5%. These techniques faced a major challenge of irrelevant feature extraction that were tried to be resolved through feature selection techniques [26].

Visual inspection of crops was carried out by farmers and agriculture experts. This evaluation process is exhausting, time-consuming, and highly subjective. The development of computer vision systems to identify, recognize, and classify disease-affected crops will keep humans out of the equation, allowing for unbiased, accurate disease-infection decisions [1]. An automatic classification system consists of various steps as mentioned above. Preprocessing is an important step, the aim is to remove noise and improve the quality of original images that later helps in important feature extraction. The extracted features from the refined images are used for the training of deep learning models that are further employed for feature extraction and classification. The key problems which are considered in this work are (i) training a deep learning model on an imbalanced dataset gives the high priority in the prediction to higher numbers of sample class; (ii) disease spots and background objects differ in appearance; (iii) changes in the shape, color, texture, and origin of the disease; (iv) irrelevant and redundant features extraction, and (v) choosing the superlative features for the classification.

In this article, our major focus is to design an automated computerized method for cucumber leaf diseases recognition using deep learning and Entropy-ELM-based best feature selection. The recent methods focused on the infected region identification and then employed for feature extraction; however, the error in the identification step misleads the irrelevant feature extraction that later reduces the classification accuracy. Our major contributions are:

(i)    Four mathematical functions such as horizontal flip, vertical flip, rotate 45, and rotation 60 are implemented for the sake of data augmentation. Later, four deep learning models are fine-tuned and trained on the augmented dataset.

(ii)   Deep learning features are extracted from the average pooling layer instead of the fully connected layer. The extracted deep features are passed to the Softmax classifier and compared the accuracy. Based on the accuracy value, the Densenet201 fine-tuned model is selected for the rest of the process. Moreover, all fine-tuned model features are fused using a new parallel approach.

(iii)  An Entropy-ELM based best feature selection technique is proposed. The proposed technique is applied on both the Densenet201 feature vector and fused vector, that later serially fused for the final classification.

(iv)  To determine which step of the proposed framework is better performed, a comparison is made between all hidden steps.

The rest of the manuscript is organized as follows: a proposed methodology that includes augmentation of the dataset, deep learning-based feature extraction, and Entropy-

ELM-based best feature selection, is presented in Section 2. Results are discussed in Section 3 with the help of tables and graphs. Finally, the conclusion of the manuscript is given in Section 4.

## 2. Proposed Methodology

In this work, an automated framework is proposed for cucumber leaf diseases recognition using deep learning and Entropy-ELM-based best feature selection. The proposed framework is illustrated in Figure 1. In this figure, it is shown that the initial augmentation step is applied to the original images by creating more training data. Then two different phases are utilized. In the first phase, four pre-trained deep models are fine-tuned and selected the best of them based on the accuracy. Features are extracted from the selected fine-tuned model and refined through the Entropy-ELM technique. In the second phase, fused the features of all four fine-tuned models and apply the Entropy-ELM technique, and finally fused with phase 1 selected feature. Finally, the fused features are classified using machine learning classifiers for the final output.

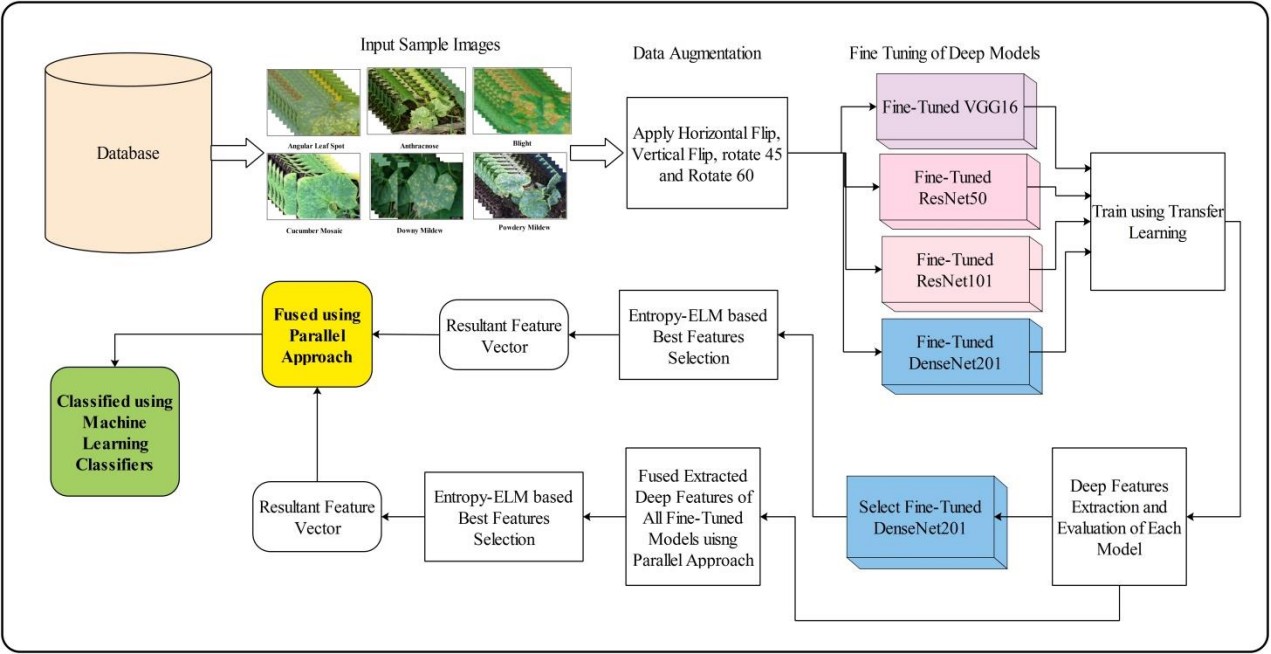

**Figure 1.** Proposed framework for cucumber leaf diseases recognition using deep learning and Entropy-ELM.

### 2.1. Dataset Collection and Augmentation

The experiments were performed on the publically available dataset named the Cucumber leaf diseases scan dataset [27]. This dataset consists of six different diseases such as anthracnose, powdery mildew, downy mildew, angular spot, mosaic, and blight. A sample of images are illustrated in Figure 2. Each class has 100 to 150 images originally that are not enough to train a deep learning model. Therefore, we design a simple algorithm (Algorithm 1) for data augmentation that includes four operations such as horizontal flip, vertical flip, rotate 45, and rotate 60. This algorithm is applied to each cucumber disease class and increases the number of images to 2000 in each class. In the later steps, this augmented dataset is utilized for the training of deep models.

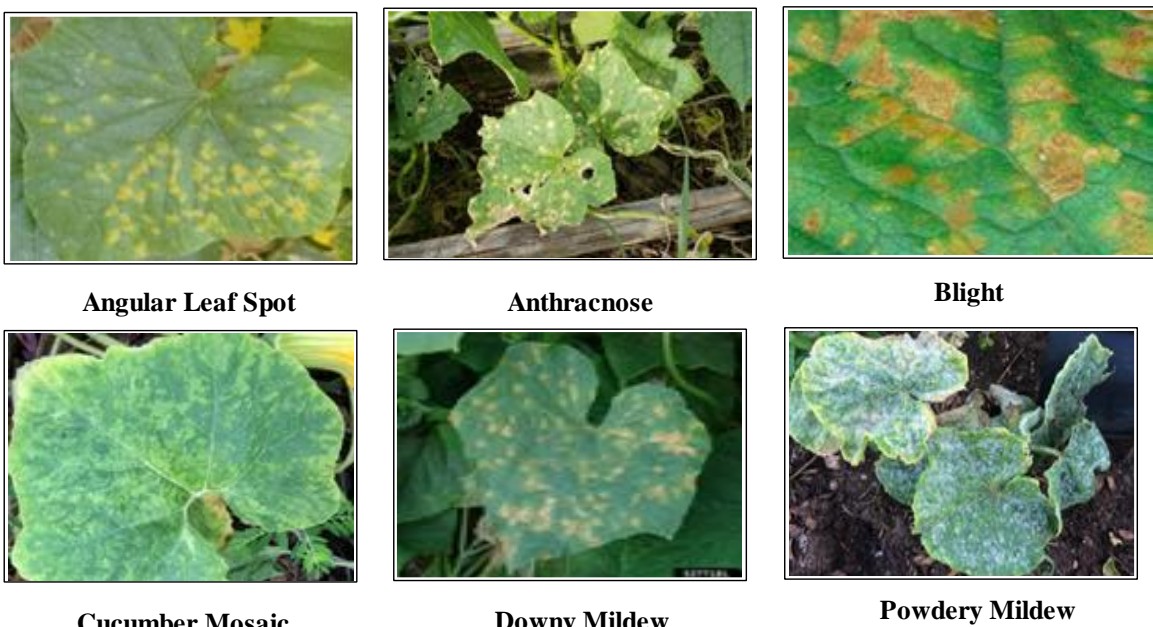

**Figure 2.** Sample images of Cucumber leaf diseases.

---

**Algorithm 1: (Data Augmentation)**

---

Step 1: Input Original Database
Step 2: Consider First Disease Class
Step 3: Count Images of Step 2 (Selected Disease Class)
Step 4: For i = 1 to Total Images of each Class

- Horizontal Flip and Image Write
- Vertical Flip and Image Write
- Rotate 45 and Image Write
- Rotate 60 and Image Write

Step 5: Repeat Step 2, 3, and 4 for the Rest of the Disease Classes
        End

---

### 2.2. Deep Learning Architecture

Four deep learning pre-trained models are employed in this work for feature extraction. The selected models are—VGG16, ResNet50, ResNet101, and DenseNet201. As mentioned in Figure 1, all selected models are initially fine-tuned and then trained through transfer learning using an augmented dataset. A brief description of each deep model is given below.

VGG16 [28] is a pre-trained model that was created by the Visual Geometry Group. This group is a combination of students and teachers focused on Computer Vision at Oxford University. This model is reflected to be one of the best computer vision models in the world. A unique feature of VGG16 is that rather than having numerous hyper-parameters it concentrates on having used identical PL and MPL of 2 × 2 filters of stride 2 and CL of 3 × 3 filters with a stride 1. VGG16 continues the same organization containing Convolutional and Maxpool Layers continuously during the course of the entire structural design. In the end, VGG has 2 Fully Connected Layers afterward a Softmax to output. Due to the fact that the VGG16 has 16 layers with weights, it has the name VGG16. This model was originally trained on an ImageNet dataset having 1000 object classes. The prediction of this model was done by the Softmax layer, defined as:

$$\Theta \ = \ w0x0 + w1x1 + \ldots + wkxk = \ \textstyle\sum_{i=0}^{k} w_i x_i = w^T x \tag{1}$$

ResNet [29] also known as a Deep Residual Network, have proved to perform with great accuracy and efficiency with a Deep Framework and to create an extra straight pathway for the transmission of data through the network. Within such Deep Systems, the deprivation issue arises because of the rise of Network Layers and the precision begins to dilute which results in its reduction quickly. Backpropagation does not come across the Vanishing Gradient problem when working with RESNET. There are some "Shortcut Connections" that a Residual Network has which are to be equivalent to a regular Convolutional Layer which aids the network to comprehend the Global Features. Then an input *x* has to be added to the output layer by adding the Shortcut connection, afterward some weight layers below. After the application of these Shortcut Connections, they permitted the network by avoiding the layers which were not beneficial while training. Hence, the output came in an ideal modification of the number of layers to perform rapid training. Mathematical, the output of *H* (*x*) can be expressed as

$$H(x) = F(x) - x \tag{2}$$

A type of Residual Mapping is used to train the weight layers which is expressed as,

$$F(x) = H(x) - x \tag{3}$$

The above-mentioned function $F(x)$ signifies stacked nonlinear weight layers. Several properties of ResNet50 include the fact that it has 64 kernels including $7 \times 7$ Convolutional layers. It also includes 16 residual blocks. There are 23 million trainable parameters.

ResNet101 model utilizes Residual links that the angles can stream straightforwardly over to hinder the slopes to get 0 after the utilization of Chain Rule. There are 104 convolutional layers altogether in ResNet101. Alongside, it comprises 33 squares of layers altogether and 29 of these squares utilize past squares yield straightforwardly which is characterized as leftover associations above. Hence the above-mentioned residuals were using such main Operand of Summation (OOS) administrator towards the termination of every square to obtain the contribution of the accompanying squares. Leftover 4 squares get the past square's yield and apply it to a CL with a channel size of $1 \times 1$ and a step of 1 after a clump standardization layer, which performs standardization activity and the resultant yield is shipped off the summation administrator at the yield of that block. Mathematically, this model working is defined as follows:

$$u(x, t+1) = u(x,t) + w(x,t) * u(x,t) \tag{4}$$

$$\widehat{T}_x = \frac{1}{2}\sigma^2 \frac{\partial^2}{\partial x^2} + b\frac{\partial}{\partial x} + c \Leftrightarrow \widehat{T}_p = -\frac{1}{2}\sigma^2 p^2 + ibp + c \tag{5}$$

$$\widehat{T}_p \widetilde{u}(p,t) = \frac{d}{dt}\widetilde{u}(p,t) \tag{6}$$

$$\widetilde{u}(p,t) = e^{\widehat{T}_p t}\widetilde{u}(p,\,0) \tag{7}$$

$$\widetilde{u}(p,t) \approx (1 + \widehat{T}_p t)\widetilde{u}(p,\,0) \tag{8}$$

Densenet-201 [30] is a convolutional neural network that is 201 layers deep. In this model, each layer gets feature maps from all preceding layers, the network can be thinner and more compact, resulting in fewer channels. The extra number of channels for each layer is the growth rate k. As a result, it has better computational and memory efficiency. The transition layers between two contiguous dense blocks are 11 Conv followed by 22 average pooling. Within the dense block, feature map sizes are uniform, allowing them to be readily concatenated. A global average pooling is done after the last dense block, and then a softmax classifier is added. The error signal can be transmitted more directly to earlier levels. As previous layers can get direct supervision from the final classification layer, this is a form of implicit deep supervision.

### 2.3. Transfer Learning Based Feature Extraction

Transfer learning (TL) is a process of reusing a pre-trained model for a new task [31], as illustrated in Figure 3. The ImageNet dataset was used as a source dataset of the pre-trained model. The pre-trained model is fine-tuned and transfer knowledge through the TL concept. In the last, the new fine-tuned model is trained on the augmented cucumber dataset that is utilized for further feature extraction. The features are extracted from the deep layers like FC7 for VGG, Average Pool for ResNet50, ResNet101, and Densenet201. Several hyperparameters are employed during the training process such as 0.0001 learning rate, max epochs are 200, the mini-batch size is 16, and the activation function is sigmoid.

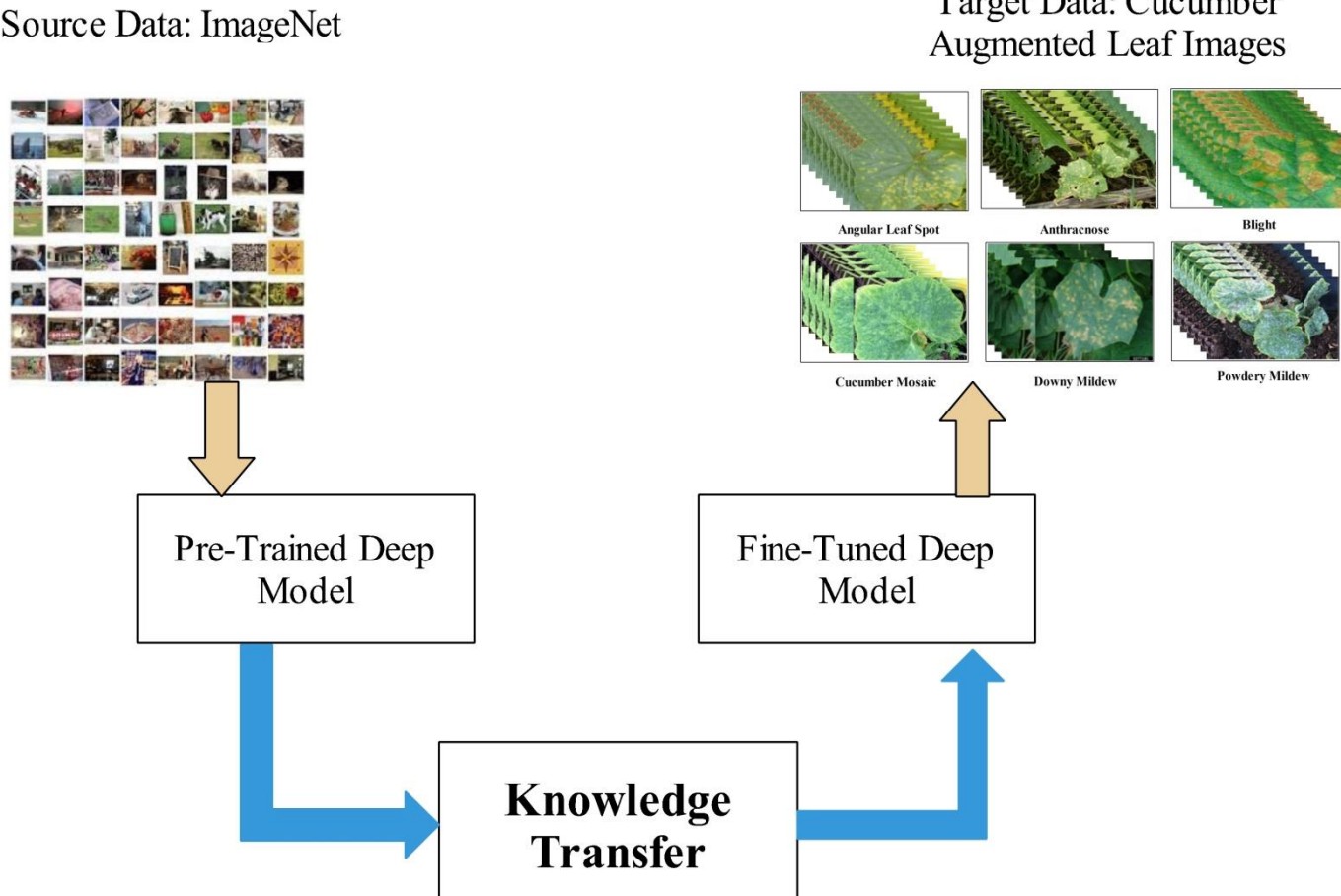

**Figure 3.** Transfer learning-based training of deep models for cucumber leaf diseases recognition.

### 2.4. Entropy-ELM Based Features Selection and Parallel Fusion

Feature selection is an important and hot research topic nowadays [32]. The main purpose of feature selection is to increase the system accuracy and minimize the computational time by focusing on the selection of the most important features [33]. In this work, a new technique is proposed named Entropy-ELM for the best feature selection. This proposed technique worked in the following steps: (i) compute the entropy of input vector; (ii) based on the entropy value, a threshold function is employed that return two vectors—fulfill the threshold value (selected) and not-selected; (iii) ELM [34] employed as a fitness function and selected threshold passed features are utilized as an input. Mathematically, the entropy formulation is defined as follows:

$$H_1 = -\sum_{k=1}^{G} P_k Id(P_k) \tag{9}$$

$$H_{diff,1} = - \int_0^1 h_1(w)1d[h_1(w)]dw. \tag{10}$$

$$H_1 = H_{diff,1}+\mathrm{Id}(\mathrm{G}) = H_{diff,1} + H_{1,max} \tag{11}$$

$$\iint_{whole\ image} [\Delta w(w,y)]^k dxdy \propto \int_{-1}^1 (\Delta w)^k h_d(\Delta w)d\Delta w = M_k \tag{12}$$

$$T = \left\{ \begin{array}{ll} Sel(k) & for\ Features(i)\ \geq H_1 \\ NotSelec\ (l) & for\ Features(i)\ < H_1 \end{array} \right. \tag{13}$$

The detail of this selection process is given in Algorithm 2.

---

**Algorithm 2: (Entropy-ELM)**

---

Step 1: Input Feature Vector $N \times K$ // $K$ is the length of features
Step 2: For i = 1 to N
Step 3: Computer Entropy through Equations (9)–(12)
Step 4: Define Threshold Function as Equation (13)
Step 5: Check Fitness through ELM
Step 6: Evaluate the Accuracy
Step 7: Repeat Step 2–6, until accuracy on the top side
        **End**
Selected Feature Vector

---

Finally, the parallel fusion approach is opted to get the fused feature vector. This approach is based on the following three steps. In the first step, get the maximum length feature vector. As we have two feature vectors $X$ and $X_1$, where the length of vectors is $N \times K$ and $N \times K_1$, respectively. In the second step, compute the entropy value and perform padding for the lower size feature vector. In the third step, correlation is computed among $K$ and $K_1$ features for the final fusion. The fused vector is finally utilized for the classification through supervised learning classifiers.

$$Fusion = \psi(K, K_1) \tag{14}$$

where $K$ and $K_1 \in X$ and $X_1$

### 3. Experimental Results

The proposed framework is evaluated on the selected cucumber dataset having a ratio of 70:15:15 which means that 70% of the images are utilized to train the model, whereas the 15% for testing and 15% for validation. We combined the testing and validation images and performed testing (30%). All the experimental results are computed with K-Fold cross-validation, whereas the value of K is 10. Several classifiers are implemented as discussed in Table 1. The performance of each classifier is computed through several measures such as recall rate, precision rate, F1-Score, accuracy, and time. The entire framework simulations are conducted on Simulink MATLAB2021a using a Personal Desktop.

**Table 1.** Brief description of selected classifiers.

| Classifiers | Details |
|---|---|
| LSVM | Kernel scale: Automatic, Box constraint level: 1<br>Multiclass method: One-vs-One |
| QSVM | Kernel scale: Automatic, Box constraint level: 1<br>Multiclass method: One-vs-One |
| CSVM | Kernel scale: Automatic, Box constraint level: 1<br>Multiclass method: One-vs-One |
| MGSVM | Kernel scale: 45, Box constraint level: 1<br>Multiclass method: One-vs-One |
| FKNN | No of neighbor: 10, Distance matric: Euclidean<br>Distance weight: Equal |
| Subspace_KNN | Learner type: Nearest neighbors, No of learners: 30<br>Subspace dimensions: 1024 |
| Weighted_KNN | No of neighbor: 10, Distance matric: Euclidean<br>Distance weight: Squared inverse |
| Cosine_KNN | No of neighbor: 10, Distance matric: cosine<br>Distance weight: Equal |
| Cubic_KNN | No of neighbor: 10, Distance matric: Minkowsi (cubic)<br>Distance weight: Equal |
| Medium_KNN | No of neighbor: 10, Distance matric: Euclidean<br>Distance weight: Equal |

### 3.1. Results

The detailed experimental process of the proposed framework is conducted in this section. The results are computed using the following steps: (i) classification using originally collected dataset on fine-tuned pre-trained models; (ii) classification using augmented dataset on fine-tuned deep models and select the best deep model for the further processing; (iii) best deep model features are refined using a new technique name Entropy-ELM; (iv) fusion of fine-tuned deep model features (augmented dataset), and (v) fused both step features using a parallel approach

### 3.2. Results on Original Cucumber Dataset

The results of the proposed method on the original cucumber dataset are given in Table 2. In this table, accuracy is computed for each fine-tuned deep model using the original dataset. Fine-tuned VGG16 (F-VGG16) obtained the maximum accuracy of 56.9% on the MG SVM classifier. The fine-tuned ResNet50 and ResNet101 obtained the best accuracy of 58.7 and 55.1% on Cubic SVM and Quadratic SVM, respectively. The fine-tuned Densenet201 deep model obtained an accuracy of 61.9% on Quadratic SVM. Based on these results, it is noticed that the originally collected dataset have several issues like imbalancing and short training data. Using these data, the fine-tuned Densenet201 gives better results for all classifiers.

**Table 2.** Classification results on originally selected cucumber dataset without data augmentation step for several fine-tuned deep learning models.

| Classifier | F-VGG16 | F-ResNet50 | F-ResNet101 | F-DenseNet201 |
|---|---|---|---|---|
| Cubic SVM | 55.6 | 58.7 | 54.2 | 61.8 |
| Quadratic SVM | 55.1 | 54.2 | 55.1 | 61.9 |
| MG SVM | 56.9 | 48 | 50.7 | 60.4 |
| Fine KNN | 50.7 | 48.9 | 49.8 | 52 |
| Linear SVM | 51.6 | 53.3 | 51.1 | 58.4 |
| ESD | 24.9 | 47.1 | 46.7 | 56.4 |
| ES KNN | 50.7 | 55.1 | 53.8 | 51 |
| WKNN | 51.6 | 47.6 | 40.9 | 49.5 |
| Cosine KNN | 47.6 | 50.2 | 43.6 | 49 |
| Medium KNN | 45.3 | 45.3 | 36.6 | 49.5 |

*3.3. Results on Augmented Cucumber Dataset*

Experimental results of fine-tuned VGG16 pre-trained model after augmentation are given in Table 3. The best-obtained accuracy is 93.8% on Cubic SVM, whereas the recall rate and precision rates are 93.84 and 93.92%, respectively. The second best-obtained accuracy is 93.6%, which was accomplished on Quadratic SVM, whereas the recall rate and precision rates are 93.66 and 93.72%, correspondingly. The execution time of Linear SVM is better than the rest of the classifiers.

**Table 3.** Classification results of fine-tuned VGG16 deep model after data augmentation.

| Classifier | Recall Rate (%) | Precision Rate (%) | Accuracy (%) | FNR (%) | F1 Score (%) | Time (Sec) |
|---|---|---|---|---|---|---|
| Cubic SVM | 93.84 | 93.92 | 93.8 | 6.16 | 93.88 | 300 |
| Quadratic SVM | 93.66 | 93.72 | 93.6 | 6.34 | 93.69 | 242 |
| MG SVM | 91.78 | 92.04 | 91.8 | 8.22 | 91.91 | 463 |
| Fine KNN | 88.54 | 88.54 | 88.5 | 11.46 | 88.54 | 562 |
| Linear SVM | 90.56 | 90.88 | 90.6 | 9.44 | 90.72 | 188 |
| ESD | 92.98 | 93 | 93 | 7.02 | 92.99 | 1526 |
| ES KNN | 88.72 | 88.74 | 88.7 | 11.28 | 88.73 | 1550 |
| WKNN | 86.22 | 86.46 | 86.2 | 13.78 | 86.34 | 1038 |
| Cosine KNN | 80.8 | 81.36 | 80.8 | 19.20 | 81.08 | 648 |
| Medium KNN | 79.44 | 80.9 | 79.5 | 20.56 | 80.16 | 589 |

The classification accuracy of fine-tuned ResNet50 on the augmented dataset is given in Table 4. This table presents the highest obtained accuracy on Cubic SVM of 94.6%, whereas the recall and precision rates are 94.36 and 94.46%, correspondingly. The second top accuracy is 94.4% obtained on Quadratic SVM, whereas the recall and precision rates are 94.26 and 94.36%, respectively. Similar to fine-tuned VGG16, the Quadratic SVM executed fast than the rest of the classifiers.

**Table 4.** Classification results of fine-tuned ResNet50 deep model after data augmentation.

| Classifier | Recall Rate (%) | Precision Rate (%) | Accuracy (%) | FNR (%) | F1 Score (%) | Time (Sec) |
|---|---|---|---|---|---|---|
| Cubic SVM | 94.36 | 94.46 | 94.6 | 5.64 | 94.41 | 964 |
| Quadratic SVM | 94.26 | 94.36 | 94.4 | 5.44 | 94.61 | 387 |
| MG SVM | 91.38 | 91.7 | 91.5 | 8.62 | 91.54 | 1169 |
| Fine KNN | 86.6 | 86.96 | 86.6 | 13.40 | 86.78 | 681 |
| Linear SVM | 91.12 | 91.48 | 91 | 8.88 | 91.30 | 657 |
| ESD | 90.22 | 90.5 | 90.5 | 9.78 | 90.36 | 655 |
| ES KNN | 91.86 | 91.6 | 91.7 | 8.14 | 91.73 | 600 |
| WKNN | 79.66 | 81.44 | 78 | 20.34 | 80.54 | 748 |
| Cosine KNN | 82.38 | 82.72 | 82.4 | 17.62 | 82.55 | 539 |
| Medium KNN | 72.64 | 76.6 | 72.6 | 27.36 | 74.57 | 392 |

Experimental results of fine-tuned ResNet101 pre-trained model are given in Table 5. The best-obtained accuracy of 97.7% was accomplished on Cubic SVM. The recall and precision rates are 97.7 and 97.7%, correspondingly. The second best-obtained accuracy is 97.2% on Quadratic SVM. The recall and precision rates are 97.24 and 97.32%, correspondingly. In this experiment, the Linear SVM was executed fast than the rest of the selected classifiers.

**Table 5.** Classification results of fine-tuned ResNet101 deep model after data augmentation.

| Classifier | Recall Rate (%) | Precision Rate (%) | Accuracy (%) | FNR (%) | F1 Score (%) | Time (Sec) |
|---|---|---|---|---|---|---|
| Cubic SVM | 97.7 | 97.76 | 97.7 | 2.30 | 97.7 | 608 |
| Quadratic SVM | 97.24 | 97.32 | 97.2 | 2.76 | 97.2 | 574 |
| MG SVM | 94.64 | 94.7 | 94.6 | 5.36 | 94.6 | 1086 |
| Fine KNN | 94.42 | 94.44 | 94.4 | 5.58 | 94.4 | 1285 |
| Linear SVM | 94.32 | 94.62 | 94.3 | 5.68 | 94.4 | 513 |
| ESD | 95.82 | 96.68 | 95.8 | 4.18 | 96.2 | 2394 |
| ES KNN | 96.36 | 96.26 | 96.3 | 3.64 | 96.3 | 4072 |
| WKNN | 92.16 | 92.48 | 92.3 | 7.84 | 92.3 | 1501 |
| Cosine KNN | 86.46 | 86.84 | 86.5 | 13.54 | 86.6 | 1404 |
| Medium KNN | 83.44 | 84.44 | 83.5 | 16.56 | 83.93 | 1342 |

The classification results of fine-tuned Densenet201 pre-trained model are given in Table 6. In this table, the obtained best accuracy is 98.4% on Cubic SVM. Moreover, the recall and precision rates are 98.44 and 98.5%, correspondingly. Figure 4 illustrated the confusion matrix of Cubic SVM that was utilized for the verification of recall rate. The second best-obtained accuracy is 97.4%, which was accomplished on Quadratic SVM. The computation time of each classifier is also noted and the minimum time is 302 (sec) for LSVM. At the first step comparison among without augmented and augmented datasets, it is noted that the accuracy obtained on the augmented dataset is significantly better. In the second step comparison, it is noted that the fine-tuned DenseNet201 model achieved better results than VGG16, ResNet50, and ResNet101. Based on this analysis, the fine-tuned DenseNet201 is selected for the rest of the experiments.

**Table 6.** Classification results of fine-tuned Densenet201 deep model after data augmentation.

| Classifier | Recall Rate (%) | Precision Rate (%) | Accuracy (%) | FNR (%) | F1 Score (%) | Time (Sec) |
|---|---|---|---|---|---|---|
| Cubic SVM | 98.44 | 98.5 | 98.4 | 1.56 | 98.47 | 355 |
| Quadratic SVM | 97.4 | 97.46 | 97.4 | 2.60 | 97.43 | 330 |
| MG SVM | 95.32 | 95.62 | 95.4 | 4.68 | 95.47 | 623 |
| Fine KNN | 93.42 | 93.42 | 93.4 | 6.58 | 93.42 | 734 |
| Linear SVM | 92.62 | 93.16 | 92.7 | 7.38 | 92.89 | 302 |
| ESD | 96.62 | 96.6 | 96.6 | 3.38 | 96.61 | 1495 |
| ES KNN | 94.1 | 94.08 | 94.1 | 5.90 | 94.09 | 1923 |
| WKNN | 92.26 | 92.4 | 92.3 | 7.74 | 92.33 | 927 |
| Cosine KNN | 85.44 | 85.94 | 85.3 | 14.56 | 85.69 | 807 |
| Medium KNN | 85.9 | 86.9 | 85.8 | 14.10 | 86.40 | 764 |

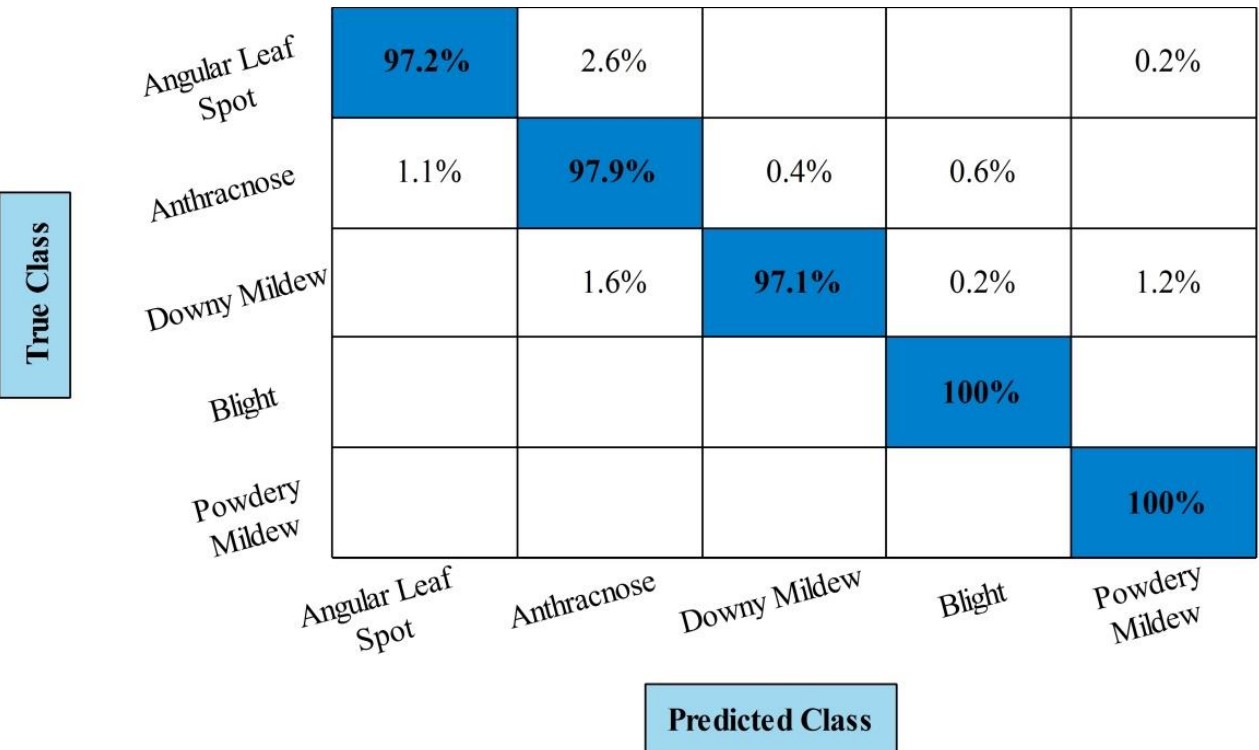

**Figure 4.** Confusion matrix-based representation of Cubic SVM accuracy.

The fine-tuned deep learning model is selected based on the better accuracy and applied proposed Entropy-ELM feature selection technique. The results are given in Table 7. This presents the best accuracy of 98% on Cubic SVM. The other computed measures are the recall rate which is 98.02, the precision rate at 97.98, and the F1-Score at 98%. The recall rate of Cubic SVM can be also verified through a confusion matrix, illustrated in Figure 5. Compared to the results given in Table 6, it is noted that the accuracy is a bit reduced but on the other side, a huge change occurred in the computation time. The time is also plotted in Figure 6 (FDenseNet201 and Dense Entropy-ELM).

**Table 7.** Classification results using proposed Entropy-ELM selection approach.

| Classifier | Recall Rate (%) | Precision Rate (%) | Accuracy (%) | FNR (%) | F1 Score (%) | Time (Sec) |
|---|---|---|---|---|---|---|
| Cubic SVM | 98.02 | 97.98 | 98.0 | 1.98 | 98.00 | 116 |
| Quadratic SVM | 96.54 | 96.62 | 96.5 | 3.46 | 96.58 | 135 |
| MG SVM | 84.2 | 81.4 | 84.0 | 15.80 | 82.78 | 197 |
| Fine KNN | 94.82 | 94.84 | 94.8 | 5.18 | 94.83 | 217 |
| Linear SVM | 82.82 | 83.14 | 82.8 | 17.18 | 82.98 | 133 |
| ESD | 93.3 | 93.78 | 93.0 | 6.70 | 93.54 | 201 |
| ES KNN | 93.16 | 92.2 | 93.0 | 6.84 | 92.68 | 405 |
| WKNN | 94.22 | 94.28 | 94.2 | 5.78 | 94.25 | 384 |
| Cosine KNN | 88.16 | 88.92 | 88.4 | 11.84 | 88.54 | 93 |
| Medium KNN | 85.78 | 86.8 | 85.8 | 14.22 | 86.29 | 204 |

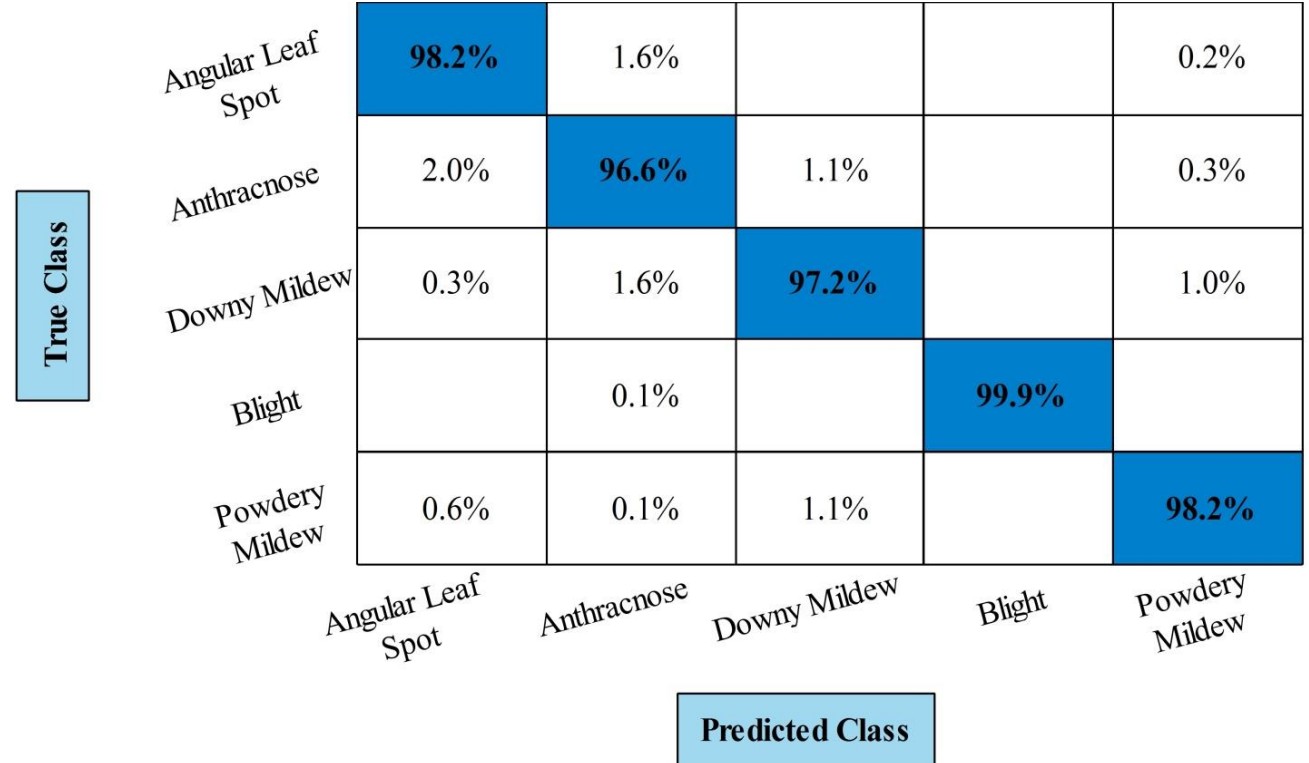

**Figure 5.** Confusion matrix of Cubic SVM after employing feature selection technique.

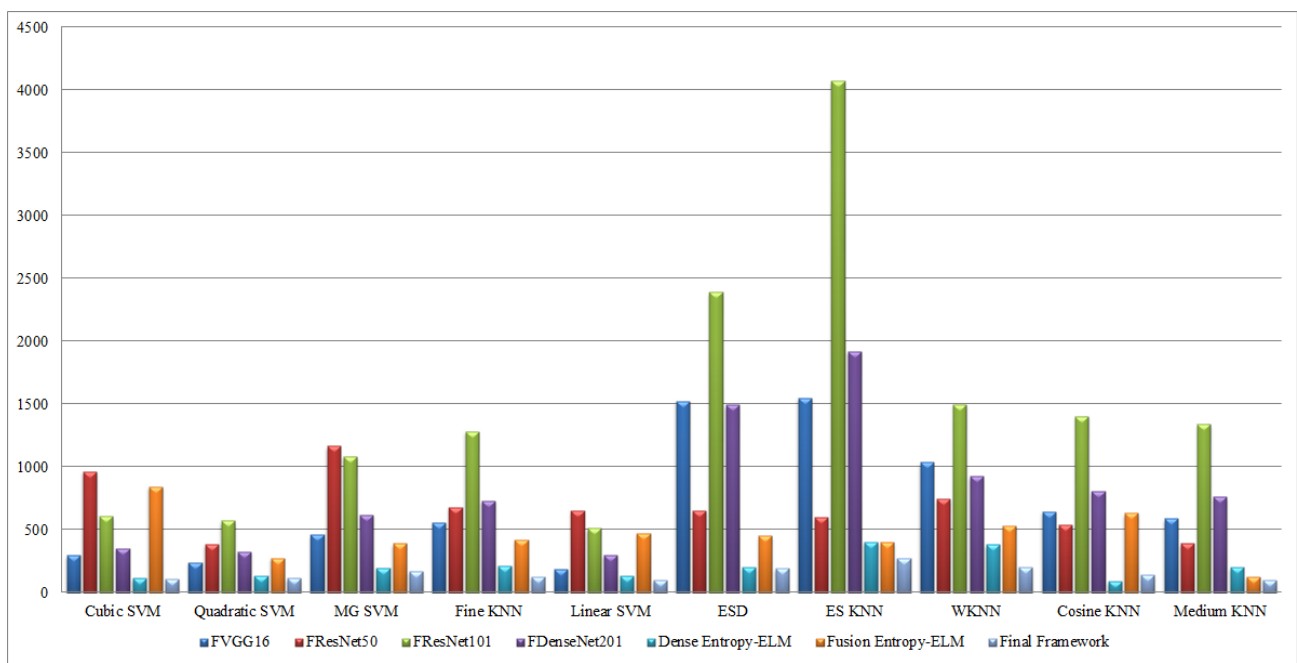

**Figure 6.** Comparison of all experiments in terms of testing time.

After the selection of the best dense features, in the next step all fine-tuned deep model features are fused using the proposed parallel approach. The results of this experiment are given in Table 8. The best-noted accuracy in this table is 98.2% on Cubic SVM. The recall and precision rates are 97.92 and 98.12%, respectively. Figure 7 illustrated the confusion matrix that can be utilized for the verification of the recall rate. The time of each classifier is also noted and plotted in Figure 6 (Fusion Entropy-ELM). In comparison with the results of Tables 6 and 7, it is noted that the overall accuracy is improved but the time is more increased than in the Dense Entropy-ELM step.

**Table 8.** Classification results using proposed parallel features fusion and Entropy-ELM selection of all pre-trained deep models using augmented dataset.

| Classifier | Recall Rate (%) | Precision Rate (%) | Accuracy (%) | FNR (%) | F1 Score (%) | Time (Sec) |
|---|---|---|---|---|---|---|
| Cubic SVM | 97.92 | 98.12 | 98.2 | 2.08 | 97.02 | 847 |
| Quadratic SVM | 97.74 | 97.38 | 97.7 | 2.26 | 97.56 | 277 |
| MG SVM | 94.84 | 95.06 | 94.8 | 5.16 | 94.95 | 392 |
| Fine KNN | 94.88 | 94.9 | 94.9 | 5.12 | 94.89 | 422 |
| Linear SVM | 94.76 | 95 | 94.8 | 5.24 | 94.88 | 475 |
| ESD | 96.26 | 96.28 | 96.3 | 3.74 | 96.27 | 454 |
| ES KNN | 96.66 | 96.66 | 96.7 | 3.34 | 96.66 | 400 |
| WKNN | 92.78 | 92.86 | 92.8 | 7.22 | 92.82 | 534 |
| Cosine KNN | 86.78 | 81.32 | 86.8 | 13.22 | 83.96 | 635 |
| Medium KNN | 84.82 | 85.6 | 84.8 | 15.18 | 85.21 | 129 |

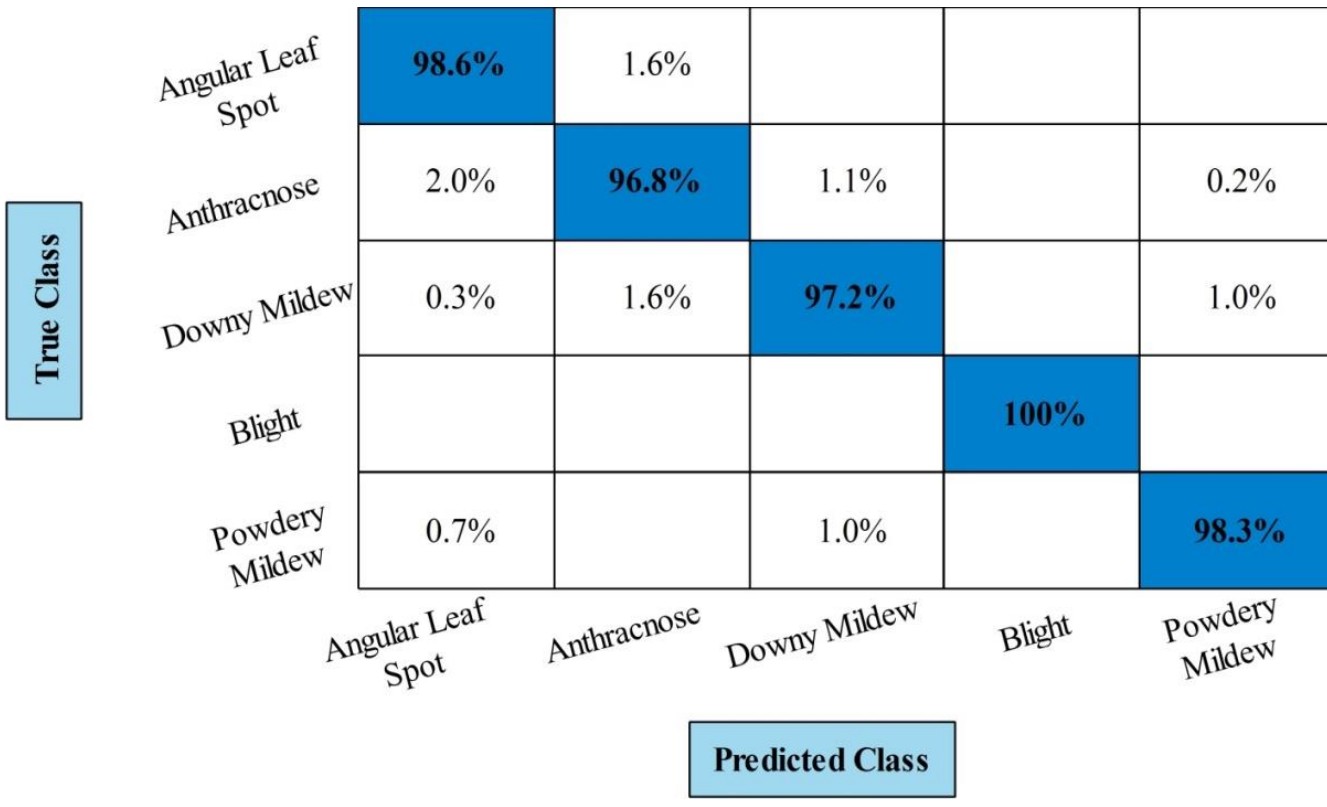

**Figure 7.** Confusion matrix of Cubic SVM after parallel fusion of all selected pre-trained deep features and Entropy-ELM selection.

Finally, the features of Dense Entropy-ELM and Fusion Entropy-ELM are fused using the proposed parallel approach, and the results are given in Table 9. This table presents the best-obtained accuracy of 98.50% on Cubic SVM. The noted precision rate is 98.30, recall rate is 98.36 and F1-Score is 98.48%, respectively. The second best-noted accuracy is 97.5% on Quadratic SVM. The recall rate of Cubic SVM can be verified through a confusion matrix plotted in Figure 8. This figure shows the correct prediction rate of each class in the diagonal. Compared to the results of this experiment with all previous experiments, it is clearly noted that the accuracy is improved and computational time is significantly reduced.

**Table 9.** Proposed framework classification results using augmented dataset.

| Classifier | Recall Rate (%) | Precision Rate (%) | Accuracy (%) | FNR (%) | F1 Score (%) | Time (Sec) |
|---|---|---|---|---|---|---|
| Cubic SVM | 98.36 | 98.3 | 98.5 | 1.74 | 98.48 | 111 |
| Quadratic SVM | 98.1 | 97.5 | 97.5 | 1.90 | 97.80 | 117 |
| MG SVM | 95.74 | 96.06 | 95.8 | 4.26 | 95.90 | 175 |
| Fine KNN | 94.42 | 94.42 | 94.4 | 5.58 | 94.42 | 130 |
| Linear SVM | 93.06 | 93.68 | 93.2 | 6.94 | 93.37 | 103 |
| ESD | 96.36 | 96.42 | 96.4 | 3.64 | 96.39 | 196 |
| ES KNN | 94.82 | 94.8 | 94.8 | 5.18 | 94.81 | 277 |
| WKNN | 92.2 | 92.56 | 92.2 | 7.80 | 92.38 | 201 |
| Cosine KNN | 85.48 | 85.78 | 85.4 | 14.52 | 85.63 | 142 |
| Medium KNN | 85.58 | 84.36 | 85.5 | 14.42 | 84.97 | 104 |

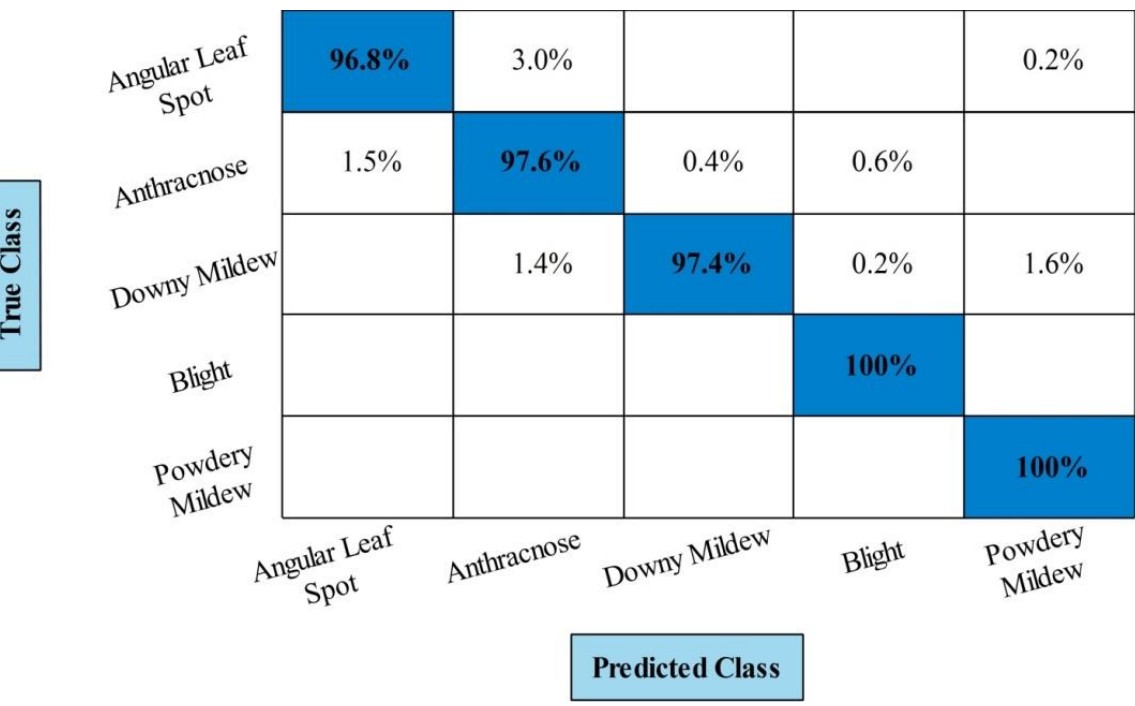

**Figure 8.** Confusion matrix of proposed framework of cucumber leaf diseases for Cubic SVM.

### 3.4. Discussion

Figure 1 showed the proposed framework that includes a few important steps. This figure illustrated the importance of the data augmentation step. The results without data augmentation having less accuracy than the results obtained after the data augmentation. Moreover, the selection of important features improves the accuracy that is later fused through a parallel approach. This step not only improves the classification accuracy but also reduced the computational time, as plotted in Figure 6. This figure clearly shows that the final fusion step significantly reduced the computational time than the rest of the steps on all classifiers.

In the last, the proposed framework accuracy is compared with recent SOTA techniques, as given in Table 10. The methods mentioned in this table are from the year 2017–2022. Moreover, all the methods mentioned in this table used the same leaf dataset. The recent best accuracy was 98.08% and 96.50% achieved by Khan et al. [13] and Hussain et al. [24]. The other methods such as Lin et al. [35] achieved an accuracy of 96.08% on the same dataset. The proposed framework achieved an accuracy of 98.48% that is improved than the SOTA techniques.

**Table 10.** Comparison with SOTA for cucumber leaf diseases recognition.

| Methods | Year | Accuracy (%) |
|---|---|---|
| Zhang et al. [27] | 2017 | 85.7 |
| Ma et al. [36] | 2018 | 93.4 |
| Lin et al. [35] | 2019 | 96.08 |
| Khan et al. [13] | 2020 | 98.08 |
| Zhang et al. [37] | 2021 | 90.67 |
| Jaweria et al. [7] | 2021 | 93.50 |
| Hussain et al. [24] | 2022 | 96.50 |
| **Proposed** | | 98.48 |

## 4. Conclusions

Agriculture is a hot topic of research nowadays. In agriculture, deep learning showed significant success from the last decade for the recognition of plant diseases. In this article, a deep learning and Entropy-ELM based framework is proposed for the recognition of cucumber leaf diseases. In the proposed framework, four pre-trained deep models are trained and selected one of them based on the accuracy that is later employed for the selection of best features using the proposed Entropy-Elm technique. In the opposite step, features of all pre-trained models are fused and apply the feature selection technique. In the last, features of both steps are fused and perform classification. The proposed framework is tested on an augmented cucumber leaf dataset and achieved an accuracy of 98.48%. Comparison with the existing techniques showed the proposed framework obtained improved results. From the results, it is concluded that the augmentation process improves the recognition accuracy but also increases the time that was the first limitation of this framework; therefore a feature selection technique is proposed to maintain the accuracy and reduce the computational time. Through feature selection and fusion process, important information is obtained that later improves the classification accuracy. Another limitation of this work was the reduction of a few features that were ignored during the selection process. In the future, EfficientNet deep model will be implemented and features will be refined through the Butterfly metaheuristic algorithm instead of the heuristic search approach [20]. Moreover, reinforcement learning and Graph CNN shall be applied and refined through feature selection algorithms for the better results [38–42].

**Author Contributions:** Conceptualization, M.A.K., A.A., A.K., S.A., A.B., M.M.I.C., H.-S.Y. and J.C.; methodology, M.A.K., A.A., A.K., S.A., A.B., M.M.I.C., H.-S.Y. and J.C.; software, M.A.K., A.A., A.K., S.A., A.B., M.M.I.C., H.-S.Y. and J.C.; validation, M.A.K., A.A., A.K., S.A., A.B., M.M.I.C., H.-S.Y. and J.C.; formal analysis, M.A.K., A.A., A.K., S.A., A.B., M.M.I.C., H.-S.Y. and J.C.; investigation, M.A.K., A.A., A.K., S.A., A.B., M.M.I.C., H.-S.Y. and J.C.; resources, M.A.K., A.A., A.K., S.A., A.B., M.M.I.C., H.-S.Y. and J.C.; data curation, M.A.K., A.A., A.K., S.A.,A.B., M.M.I.C., H.-S.Y. and J.C.; writing and original draft preparation, M.A.K., A.A., A.K., S.A., A.B., M.M.I.C., H.-S.Y. and J.C.; writing, review and editing, M.A.K., A.A., A.K., S.A., A.B., M.M.I.C., H.-S.Y. and J.C.; visualization, M.A.K., A.A., A.K., S.A., A.B., M.M.I.C., H.-S.Y. and J.C.; supervision, M.A.K., A.A., A.K., S.A., A.B., M.M.I.C., H.-S.Y. and J.C.; project administration, M.A.K., A.A., A.K., S.A., A.B., M.M.I.C., H.-S.Y. and J.C.; funding acquisition, M.A.K., A.A., A.K., S.A., A.B., M.M.I.C., H.-S.Y. and J.C. All authors have read and agreed to the published version of the manuscript.

**Funding:** This work was supported by the National Research Foundation of Korea (NRF) grant funded by the Korean government (Ministry of Science and ICT; MSIT) under Grant RF-2018R1A5A7059549.

**Institutional Review Board Statement:** Not applicable.

**Informed Consent Statement:** Not applicable.

**Data Availability Statement:** Not applicable.

**Conflicts of Interest:** The authors declare no conflict of interest.

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
