# Peer review of "Cucumber Leaf Diseases Recognition Using Multi Level Deep Entropy-ELM Feature Selection"

_applsci, doi:10.3390/app12020593_

Round 1

Reviewer 1 Report

The paper is presented an interesting idea and it is well written. However, I would like to propose several comments for its further improvement. 

the methodology section is well developed, but the quality of the figures needs to be improved, perhaps saving the original version with higher resolution and DPI would solve this issue.  

the discussion section is missing, such work needs an intensive discussion section for sure, by means of discussing the efficiency of methods and finding results against the similar studies and etc,  I assume some statements about machine learning methods and perhaps deep learning techniques for future studies would be efficient, here are some samples that can be taken into account for the further improvement of the paper. 

https://www.tandfonline.com/doi/abs/10.1080/15481603.2021.2000350

https://www.tandfonline.com/doi/full/10.1080/09640568.2021.2001317

Author Response

Response sheet has been attached. thanks

Reviewer 2 Report

I have made comments directly on the manuscript attached, but the introduction should be corrected. The agronomic part is not correctly referenced with appropriated litterature and the objectives of the study are not correctly presented, as well as the others studies close to this one. I've observed that any authors come from agronomic side, but cite other studies on "detection" of diseases are not the goog ressource to explain in what this study may help farmers to make decisions or undernstand how diseases affects the growth of the plants.

For this last part, there is only a brief description of the methodology used in the litterature and what it's going to be explained in the text, but there is no factual analysis of the limits of the previous works and how the present manuscript will have an advantage compared to the previous ones.

Materials and methods and results are clear.

Author Response

Response sheet has been attached. 

Round 2

Reviewer 1 Report

Thanks for the comprehensive and prompt revise, 

Author Response

Response sheet has been added. thanks

Reviewer 2 Report

Authors have correctly modified the document and take into account my remarks.

There are still some minor corrections on english to be done. I've highlighted them in the document attached.

Author Response

(The authors gave the same response as above.)
